# Spatial Heterogeneity and Methodological Insights in Fish Community Assessment: A Case Study in Hulun Lake

**DOI:** 10.3390/biology14121678

**Published:** 2025-11-26

**Authors:** Zifang Liu, Yuetong Zhang, Yanan Pan, Zhousunxi Ma, Xin Han, Ziqi Zhou, Shuang Tian, Bingjiao Sun

**Affiliations:** 1Guangdong Provincial Key Laboratory of Water Quality Improvement and Ecological Restoration for Watersheds, School of Ecology, Environment and Resources, Guangdong University of Technology, Guangzhou 510006, China; zifang.s.liu@gmail.com; 2China National Environmental Monitoring Centre, Beijing 100012, China; 3College of Fisheries Sciences, Tianjin Agricultural University, Tianjin 300392, China

**Keywords:** fish community structure, biodiversity monitoring, ASV vs. OTU, spatial ecology

## Abstract

To protect the biodiversity of Hunlun Lake, an important nature reserve in China, we compared a molecular method—analyzing DNA left by organisms in the water (environmental DNA)—with the capture-based traditional survey methods. The eDNA method detected 2–3 times more fish species than by nets, including rare and endangered species. Both methods showed that fish communities differ in different parts of the lake. The central lake area had a unique assemblage of species. The combined approach of eDNA and capture survey can help guide more effective conservation efforts to protect the lake’s ecosystem.

## 1. Introduction

Lakes are critical reservoir freshwater biodiversity, providing ecosystem services and acting as hubs for biogeochemical cycles and species interactions [1]. Despite covering less than 1% of Earth’s surface, inland waters support over 10% of all known species, including a third of vertebrate diversity [2]. However, these systems are among the most threatened by climate change, habitat degradation, and invasive species [3].

Hulun Lake (Dalai Lake), situated at the tri-national border of China, Mongolia, and Russia, was designated a UNESCO Biosphere Reserve in 2002 under the Man and the Biosphere Programme (MAB) due to its unique grassland–wetland ecosystems and role as a critical stopover for migratory birds (UNESCO). This biodiverse region supports endemic fish communities and acts as a climate buffer, yet its ecological integrity is increasingly compromised [4]. Since the early 21st century, water levels have declined and quality has decayed rapidly due to extreme droughts, overfishing, and unregulated aquaculture [5]. These pressures disrupt ecosystem functions—eutrophication from aquaculture promotes algal blooms, and invasive species introductions risk catastrophic shifts in native fish assemblages [5]. Therefore, conservation biomonitoring actions are in critical need.

Conventional fish monitoring in Hulun Lake relies on capture-based surveys, which yield morphological data but suffer from low sensitivity for rare or invasive species, taxonomic biases, and ecological disturbance [6]. In contrast, environmental DNA (eDNA) metabarcoding provides a non-invasive, high-sensitivity alternative by detecting trace DNA in water samples, enabling comprehensive biodiversity assessments with minimal ecosystem impact. However, eDNA accuracy in Hulun Lake may be compromised by tourist activities and water quality, which can introduce false positives (e.g., DNA from resturants and recreational fishing ponds) or false negatives (e.g., PCR inhibition from organic pollutants) [7]. Additionally, while several studies suggest that eDNA metabarcoding can provide quantitative or semi-quantitative estimates of species abundance [8,9,10], conclusions across studies remain inconsistent [11]. These inconsistencies are further compounded by differences in bioinformatic pipelines (e.g., Amplicon Sequence Variant (ASV) vs. Operational Taxonomic Unit (OTU) clustering), which can influence species detection and abundance estimation [12,13,14]. While comparisons between ASV and OTU pipelines have been conducted in various aquatic ecosystems, their performance and implications for conservation have not been evaluated in the unique context of Hulun Lake.

Therefore, we integrate eDNA metabarcoding with traditional capture-based surveys to conduct the following:(i)Assess methodological performance in characterizing fish assemblages, including comparisons of alpha and beta diversity metrics recovered by ASV- and OTU-based eDNA pipelines and their correlations with traditional survey data.(ii)Identify habitat-driven community clusters and quantify spatial variability in fish diversity using both eDNA and in-net data.(iii)Explore the relationship between fish community patterns and anthropogenic stressors to inform targeted conservation strategies. By bridging molecular and traditional monitoring approaches, this study aims to improve spatial conservation planning for the Hulun Lake ecosystem.

## 2. Materials and Methods

### 2.1. Sampling

In Hulun Lake (48°97′ N, 117°43′ E; Figure 1), eDNA and capture-based surveys took place at 21 sampling sites in which 18 sites were distributed in Hulun Lake, and 3 sites were located in Wulan Nuoer Lake, connected to Hulun Lake by its tributary—Wuerxun River. These sampling sites, specified as four clusters, allow assessment of hydrological connectivity and human impact on fish assemblages by including high human impact areas—tourist facilities impact regions (*TFIR*), low human impact areas—lake centre (*Lake Centre*), and the inflow (*Inflow*) areas from the connected Wuerxun River and Wulan Nuoer Lake (*Wulan Nuoer*). From each sampling site, three biological replicates were taken separately and for each biological replicate, 1 L of surface water was filtered to concentrate eDNA. Within 12 h of collection, each sample was filtered through an 0.45 μm Sterivex-GP PES filter (SVGP01050; Merck Millipore, Darmstadt, Germany) using a peristaltic pump. An amount of 1 L of distilled water was taken into the field as a field negative control by exposing to sampling equipment. At least one field negative control was prepared per sampling day and treated identically to the other samples. The DNA-enriched filter membranes were preserved in ATL buffer (Qiagen, Hilden, Germany) and stored at −20 °C pending subsequent extraction.

Fish samples were collected using gill netting at the same sampling locations as eDNA sampling, mainly following a previous report [15]. Prior to sampling with gillnets, the sampling range and time were reported to the relevant authorities in accordance with local conservation measures. The fishing gear used in this survey included the multi-meshed gill net (2 m × 30 m, composed of 12 types of mesh sizes, ranging from 2.7 cm to 16 cm), the three-layer gill net (1 m × 50 m, with mesh sizes of 3.5 cm, 5.5 cm, and 7.5 cm), and the bottom-fixed trap net (0.5 m × 0.5 m × 15 m). Gillnets were deployed for more than 12 h after eDNA sampling between 5 and 7 pm in each sampling day, then retrieved the following morning 6–8 am. After retrieval, the catch from each net was identified into species level by local fish taxonomists. Scientific names and number of counts were record for further analysis.

### 2.2. DNA Extraction, PCR Amplification, and Illumina Sequencing

DNA was extracted from filters using the DNeasy Blood and Tissue Kit (Qiagen, Hilden, Germany). Using PCR, we amplified a ~165 bp fragment of the mitochondrial 12S gene using the Tele02 teleofish primer pair [16], which are modified versions of the MiFish-U primers [17]. These PCR primers were adapted with unique 8 mer sample-identifying barcode tags identical on both the forward and reverse primer, and incorporating 2–4 random 5′ bases to increase sequencing heterogeneity. A total of four PCRs were performed on each extracted eDNA template. Each PCR was conducted in a 20 μL volume comprising 10 μL Rapid Taq Master Mix (P222-02; Vazyme, Nanjing, China); 1 μL forward primer (5 μM); 1 μL reverse primer (5 μM); 2 μL molecular-grade water; and 6 μL eDNA template (prediluted to 20 ng/μL). Thermocycling parameters comprised polymerase activation at 95 °C for 5 min; 40 cycles of 95 °C for 15 s, 54 °C for 15 s, 72 °C for 12 s; and a final extension of 72 °C for 5 min. Alongside the extracted 75 samples, we included six filtration negative controls, one extraction negative control, and one negative no-template PCR control (Appendix A). The eDNA extractions, pre-PCR preparations, and post-PCR procedures were carried out in separate rooms. PCR products were checked by gel and then pooled and purified using the Vahta DNA Clean Beads (N411; Vazyme, Nanjing, China) following the manufacturer’s protocol. Illumina sequencing adapters were attached to the amplicons using the Vahts Universial DNA Library Prep Kit for Illumina V3 (ND607; Vazyme, Nanjing, China) following the manufacturer’s protocol. A library was prepared and then quantified using a NEBNext (E7630S; New England Biolabs, MA, USA) qPCR assay and sequenced on an Illumina Miseq using v2 (2 × 150 bp pair-end) chemistry (Illumina, Inc., San Diego, CA, USA).

### 2.3. Bioinformatic Analyses

We processed raw eDNA sequences using two bioinformatic pipelines; one is based on error detection, which generates ASVs using ‘DADA2’ plugin under QIIME 2 (v2024.10.1) [18] framework; another is based on clustering, which generates operational taxonomic units (OTUs) using OBITools (v 4.4.0) [19] pipeline. QIIME2 ASV-based workflow consisted of the following steps: (i) demultiplexing and quality control to use cutadapt [20] trim primers and assign reads to samples; (ii) denoised paired-end reads using DADA2 [21] with parameters: --p-trunc-len-f 220 --p-trunc-len-r 180 --p-max-ee 2.0, then chimeras removed via ‘consensus’ method. A denoised fasta file and ASVs table were generated after two main steps and prepared for following taxonomy assignment.

OBITools workflow followed the ‘cookbook’ (https://obitools4.metabarcoding.org/, accessed on 8 October 2024), with the following four main steps: (i) pair-end read assembly and filtering improperly joined annotations using ‘obipairing’ and ‘obigrep’; (ii) demultiplexing and sample assignment according to a barcodes file prepared as per required format using ‘obimultiplex’; (iii) dereplication—assigned reads were collapsed into unique sequences by sample using ‘obiuniq’; (iv) error filtering and chimera removal: sequence-level error filtering was conducted with ‘obiclean’, using a minimum relative abundance threshold (-r 0.1) and chimera detection (--detect-chimera). The -H option was enabled to handle homopolymer errors. Cleaned sequences were further filtered to retain only those longer than 100 bp using ‘obigrep’. A cleaned fasta file and OTU table were generated after four main steps and prepared for the following taxonomy assignment.

Taxonomic assignment for both ASVs and OTUs: taxonomic classification was performed using ‘vsearch’ (70% confidence threshold) [22], blastn (v2.11.0), and EPA-ng v0.3.8 [23] against a curated reference database to obtain the best results to species-level identification. Reads present in negative controls were used as species-specific cutoffs, and any taxon with fewer reads than observed in negative controls was excluded from downstream analyses. This procedure ensured that potential contamination did not affect community structure results. Both datasets generated from two bioinformatic modules (QIIME2 and OBITools, Appendix A) will be included in the downstream analyses.

### 2.4. Statistical Analyses

All downstream statistical analyses and visualizations were conducted in R v4.5.0 [24]. Species accumulation curves were generated using the specaccum function (vegan v2.6-10) [25] with the “random” method to assess sampling completeness across the three datasets: two derived from eDNA metabarcoding (ASV and OTU) and one from traditional capture-based surveys. Significance of species richness differences among the datasets was evaluated via one-way ANOVA (‘anova’ function, vegan v2.6-10) [26], followed by post hoc *t*-tests using the traditional survey result as a baseline.

To delineate site groupings, hierarchical clustering was performed on Hellinger-transformed [24] traditional capture-based abundance data using the ‘hclust’ function in stats v4.50 [24]. Fish community composition was compared among clusters using multivariate approaches. Jaccard dissimilarity matrices were derived from Hellinger-transformed data for all datasets. Community patterns were visualized via principal coordinate analysis (PCoA; ‘pcoa’ function, ape v5.8-1). In post hoc analysis, pair-wised permutational multivariate ANOVA (PERMANOVA) tests (999 permutations) were performed using the ‘pairwise.adonis’ function in the pairwiseAdonis v0.4.1 [27] to evaluate differences in community structure among ecological groups. Additionally, similarity percentile (SIMPER) analysis was used to identify the taxa contributing to the dissimilarity between clusters using ‘simper’ function in vegan [25].

Finally, linear regression models were used to investigate the correspondence in species abundance patterns among different methodologies. Both eDNA datasets were 4th-root and log-transformed, and in-net abundance data were transformed to frequencies using the ‘decostand’ function. These models were implemented with the ‘lm’ function [24] to account for overdispersion and heteroscedasticity.

## 3. Results

### 3.1. Alpha Diversity and Species Composition

Sequencing yielded 2,892,001 reads (QIIME2/ASV) and 2,190,909 reads (OBITools/OTU; Appendix A). After quality filtering and contamination removal (based on negative controls), 43 taxa (40 species, 3 genera) were identified across 75 samples. The ASV approach detected 40 taxa, the OTU method recovered 31 taxa, while the capture-based survey only retrieved 13 species. Notably, one critically endangered species, *Acheilognathus hypselonotus*, was found in the OTU dataset and one vulnerable species, Choi’s spiny loach (*Cobitis choii*), was found in the ASV dataset. Species accumulation curves (Appendix A) indicated that sufficient sampling effort for both eDNA and net captured the majority of species present in the ecosystem.

Most species collected by nets were found in both eDNA datasets, with the exception of silver carp (*Hypophthalmichthys molitrix*) which was absent in the OTU results, likely being clustered to bighead carp (*Hypophthalmichthys nobilis*) due to their highly genetic similarity [28,29] (Figure 2). Interestingly, several common local species—such as Prussian carp (*Carassius gibelio*), crucian carp (*Carassius carassius*), and Amur goby (*Rhinogobius brunneus*)—were not detected in net but were found in high abundance in both eDNA datasets (Appendix A; Figure 2). Comparing species recovered by two eDNA approaches, only three species—Khanka spiny bitterling (*Acanthorhodeus chankaensis*), *Acheilognathus hypselonotus*, and *Micropercops swinhonis*—were found in the OTU results but not in the ASV results. Conversely, 11 species were uniquely detected in the ASV dataset (Figure 2). Despite differences in total species richness revealed by three methods, a post hoc *t*-test revealed no significant difference in alpha diversity between the two eDNA approaches. However, both eDNA datasets showed significantly higher alpha diversity compared to the capture-based data (Figure 2).

Linear regression analysis revealed highly significant associations between species abundance metrics across capture-based and eDNA approaches (Figure 3). A significant positive correlation was observed between frequencies from nets and both the ASV and OTU reads under both log and 4th-root transformations. The OTU results demonstrated a slightly better fit than the ASV results in these models, likely due to a smaller proportion of absent species. Strong correlations were also observed between the two eDNA pipelines under both transformation methods, indicating similar performance and supporting the consistency of the alpha diversity comes.

### 3.2. Beta Diversity and Community Structure

The optimal number of clusters (k = 4) in hierarchical clustering model was determined via an elbow plot (Appendix A). This analysis delineated four distinct ecological groups: *TFIR*—four sampling sites located adjacent to tourist facilities; *Inflow* region—three sites situated next to Wuerxun River inflow; *Wulan Nuoer*—three sites within Wulan Nuoer Lake, which is hydrologically connected to Hulun Lake via the Wuerxun River; and *Lake Centre*—eleven sites situated in the central basin of Hulun Lake. Statistically significant differences (*p* < 0.05) among the four ecological clusters were observed via principal coordinate analysis (PCoA, Figure 4; Table 1) in all three datasets (net and ASV/OTU-based eDNA). Pairwise comparisons demonstrated consistent differentiation between the *Lake Centre* and other regions across all datasets. From both PCoA analysis and PERMANOVA model, the strongest dissimilarities (clear separation between clusters) were observed in the capture-based data, while the eDNA datasets revealed weaker but still significant contrasts (Table 2, Figure 4). Fish community in *TFIR* is significantly different from Wulan Nuoer, again, across all datasets. Notably, *Inflow* region and *TFIR* exhibited marginal differentiation in net (*p* < 0.05) and ASV-based eDNA data (*p* < 0.05), but not in the OTU dataset; in contrast, *Inflow* region and *Wulan Nuoer* showed significant differences in the OTU dataset, but in neither capture-based data nor ASV results, suggesting different bioinformatic approaches in eDNA studies vary in their sensitivity to fine-scale community differences.

The SIMPER analysis was performed to assess spatial variation in fish community composition among four distinct areas. Across all methodological approaches, key species such as Amur catfish (*Silurus asotus*) and *Hemiculter bleekeri* consistently accounted for a substantial proportion of community dissimilarity. Amur catfish (*S. asotus*) was particularly abundant in the Wuerxun River inflow region across all methods, indicating potential niche partitioning or habitat preferences. In contrast, *H. bleekeri*, a generalist species, played a central role in distinguishing fish communities among the four areas, although its ecological contribution varied across the three datasets (Figure 5).

The traditional netting data revealed even more pronounced species-specific differences. Yellowhead catfish (*Tachysurus fulvidraco*) was a dominant contributor in multiple comparisons due to its relatively low abundance in the *Inflow* region. Additionally, *Carassius auratus* were found in high abundance in *Wulan Nuoer* and contributed significantly to community differences relative to other areas. Predatory carp (*Chanodichthys erythropterus*) showed greater abundance near the *TFIR*. *H. bleekeri* also played a key role in differentiating the *TFIR*, the *Inflow* region, and *Wulan Nuoer*, likely due to its high abundance in the *Lake Centre* (Figure 5, Appendix A).

In the ASV-based eDNA results, generalists such as common carp (*Cyprinus carpio*), Prussian carp (*Carassius gibelio*), and *H. bleekeri* emerged as major contributors to dissimilarities. Common carp (*C. carpio*) was found in higher amounts near *TFIR*, plausibly indicating strong adaptation to anthropogenic stress. Prussian carp (*C. gibelio*) is a common species in Hulun Lake [4,30] and was found in a great abundance in eDNA results but failed to be found in nets this time. Because of its wide condition tolerance, it was found in all sampling areas with relatively higher abundance in *Wulan Nuoer* (Figure 5; Appendix A).

Similarly, in the OTU-based eDNA dataset, widely adapted species such as crucian carp (*C. carassius*), Prussian carp (*C. gibelio*), and H. bleekeri were key drivers of dissimilarity, particularly between the *TFIR* and *Wulan Nuoer*; *H. bleekeri* preferred *Wulan Nuoer* while two carps preferred *TFIR*. Notably, Amur goby (*Rhinogobius similis*) emerged as a significant contributor in *Wulan Nuoer* (*p* < 0.05). This pattern was consistent with the ASV dataset, though this native species was absent in the net, likely due to its small body size [31] (Figure 5; Appendix A).

## 4. Discussion

Our results validated eDNA as a complementary tool to traditional methods, revealing consistent species detection patterns while highlighting habitat-driven differences in fish communities. By integrating molecular and taxonomic approaches, this study provides a framework for monitoring the ecological integrity of Hulun Lake and guiding evidence-based restoration efforts.

### 4.1. Methodological Consistency

Comparisons between two eDNA clustering approaches (ASV and OTU) and the capture-based survey revealed both convergence and divergence. No significant differences in alpha diversity were observed between the ASV and OTU datasets, indicating consistency between the two bioinformatic pipelines for general diversity assessments—consistent with previous studies [32,33]. Linear regression analyses showed significant positive correlations between fish frequencies in the capture-based survey (*p* < 0.001) and both eDNA datasets, supporting the reliability of eDNA in estimating relative abundance. The significant positive correlation also supports the consensus that eDNA metabarcoding provides semi-quantitative data [11]. Notably, the OTU-based results exhibited a slightly better fit with in-net data, potentially due to fewer false positives [34].

However, discrepancies in species detection were evident. eDNA approaches outperformed capture-based survey in detecting small-bodied or rare taxa (e.g., *Cobitis choii* and *Rhinogobius similis*), highlighting their value for comprehensive biodiversity monitoring [35,36]. At the same time, the OTU pipeline’s tendency to cluster sequences with high genetic similarity likely led to the misidentification or merging of closely related species or hybrids, such as the absence of silver carp (*Hypophthalmichthys molitrix*) which was likely clustered with bighead carp (*H. nobilis*) [37]. Conversely, the ASV pipeline’s single-nucleotide resolution, while powerful, may be prone to over-splitting in this high-biomass environment, potentially detecting rare variants that could be false positives [38]. Thus, a dual-pipeline approach, as adopted here, provides complementary perspectives. It is worth noting that clustering methods alone may not fully explain these discrepancies; additional factors such as primer choice, PCR efficiency, and the thresholds used in bioinformatic processing also play critical roles [39,40,41,42].

SIMPER analyses further emphasized the complementary nature of the two methods: capture-based survey tended to detect larger, benthic species (e.g., *T. fulvidraco*), while eDNA captured a widely adapted taxa (e.g., *C. gibelio* and *H. bleekeri*), including small and pelagic species (e.g., *R. similis*) often missed by traditional surveys [42,43]. These findings suggest that while eDNA provides a more comprehensive biodiversity snapshot, combining it with traditional methods enhances detection reliability. In terms of eDNA methodology, future studies should optimize primer selection for local species, experiment design and bioinformatic pipelines to minimize false negatives, and improve cross-method consistency [7,44].

### 4.2. Spatial Heterogeneity in Fish Communities

Clear spatial structuring was evident across Hulun Lake, with four ecological zones identified and named using their location character: *TFIR* sampling sites are located adjacent to tourist facilities; *Inflow* region includes samplings sites near to Wuerxun River; *Wulan Nuoer* is hydrologically connected to Hulun Lake via the Wuerxun River; and *Lake Centre* situated in central Hulun Lake. This spatial heterogeneity aligns with known hydrological and anthropogenic influences, such as sightseeing activities in the *TFIR* zone, riverine inputs shaping community composition near the *Inflow* of Wuerxun River, and the geographic division of Wulan Nuoer Lake.

The SIMPER analysis further highlighted key species driving these differences. *H. bleekeri* was consistently a major contributor to dissimilarities, reinforcing its roles as ecological generalists with broad habitat tolerances [45]. In contrast, Amur catfish (*Silurus asotus*) showed strong associations with channels and benthic habitats, suggesting niche specialization [46].

The consistent differentiation of the *Lake Centre* from other areas across all datasets suggests that central habitats may support distinct assemblages—potentially due to their distance from shore, reduced anthropogenic pressure, or different prey availability [47]. Meanwhile, the marginal differentiation between *Inflow* region and *TFIR* in-net and OTU data (but not ASV) implies that eDNA bioinformatic pipelines vary in their sensitivity to subtle habitat gradients in complex freshwater systems. Similar observations have been reported in other lake and river studies, where clustering methods (OTU vs. ASV) yielded differing resolutions of community structure due to the influence of read denoising, chimera filtering, or clustering thresholds [12,13,14]. These findings contrast with a previous conclusion, who reported negligible pipeline effects in a more homogenous lake system [48]. Together, these results underscore the need to consider methodological choices when interpreting eDNA-based community patterns—especially in spatially heterogeneous environments. Incorporating such variation into conservation planning is essential, as different zones may require targeted management based on their distinct species assemblages.

### 4.3. Conservation and Management Implications

The detection of threatened species (*A. hypselonotus* and *C. choii*) and the identification of distinct fish community structures across Hulun Lake underscore the need for spatially explicit conservation measures [35,49]. The dominance of generalist species like crucian carp (*C. carassius*), Prussian carp (*C. gibelio*), and common carp (*C. carpio*) in disturbed areas (e.g., *TFIR*) suggests that anthropogenic activities may favour adaptable taxa [50]. In contrast, unique assemblages found in the *Lake Centre* may serve as a refuge for biodiversity, warranting prioritization in conservation planning [46].

Therefore, we recommend prioritizing the *Lake Centre* zone as a core conservation area with restricted anthropogenic disturbance. *Tourist facilities* adjacent zones require regular monitoring for invasive species from compassionate release activities, particularly generalists such as *Carassius gibelio*, studying the ecological effect of invasive and introduced species and closely monitoring invasive and generalist species to safeguard native assemblages [51]. To support this, eDNA monitoring could be integrated into regulatory frameworks to provide early warnings of invasive species spread or biodiversity loss.

When framing eDNA-based regulations, it is crucial to consider the method’s limitations. For instance, PCR inhibition from organic matter or algal blooms, which can reduce detection sensitivity, and incomplete reference databases, which can lead to false positives/negatives or misidentification of hybrids and closely related species, and so on [52]. Future applications should focus on expanding reference databases and incorporating technical replicates to improve reproducibility [52,53,54].

The complementary strengths of eDNA- and capture-based surveys advocate for integrated monitoring frameworks. eDNA is particularly useful for rapid biodiversity assessments and detecting small or rare species, while capture-based survey provides abundance data and ecological insights for larger, commercially relevant species [55,56]. By leveraging multi-method approaches and spatial community data, managers can develop more effective strategies to preserve freshwater lake biodiversity amid growing environmental challenges.

## 5. Conclusions

This study demonstrates that integrating environmental DNA (eDNA) metabarcoding with traditional trawl surveys offers a powerful, complementary approach for monitoring fish biodiversity in vulnerable ecosystems like Hulun Lake. While eDNA methods significantly outperformed capture-based survey methods in detecting species richness, especially rare, small-bodied, or endangered taxa, capture-based data provided valuable abundance information for larger species. Both methods consistently revealed significant spatial variation in fish communities across four distinct ecological zones, driven by hydrological and anthropogenic factors. The central lake emerged as a potential refuge for unique assemblages, underscoring the need for spatially targeted conservation. Although bioinformatic choices (ASV vs. OTU) influenced fine-scale community resolution, both eDNA pipelines showed strong correlation with trawl data and between themselves. We recommend adopting a combined monitoring framework to enhance detection accuracy, support evidence-based management, and ultimately contribute to the preservation of biodiversity in anthropogenically stressed freshwater systems.

## Figures and Tables

**Figure 1 biology-14-01678-f001:**
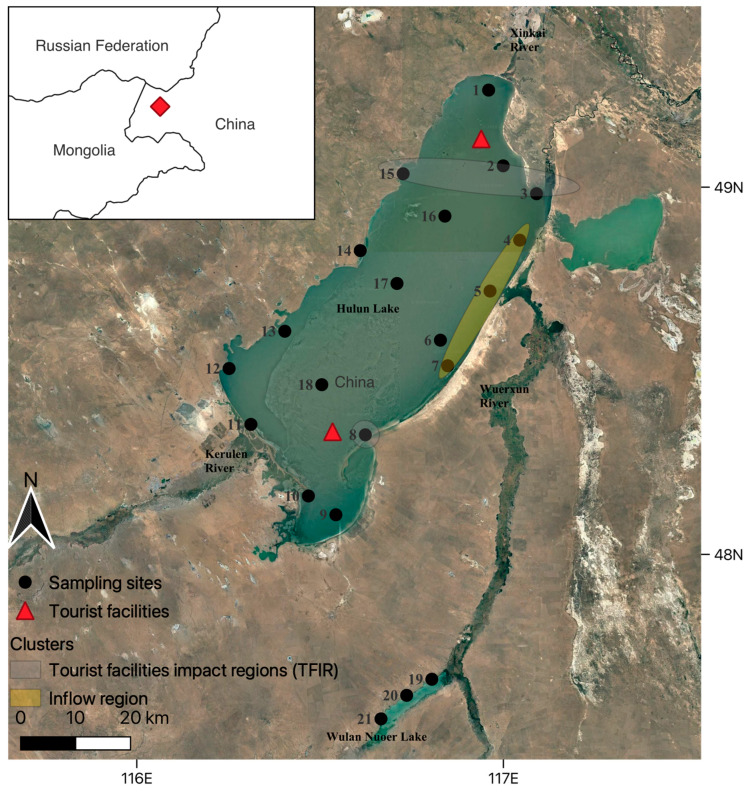
Sampling sites in the Hulun Lake wetland (48°97′ N, 117°43′ E) in Inner Mongolia, China. A total of 21 sampling sites where traditional capture-based and eDNA surveys took place.

**Figure 2 biology-14-01678-f002:**
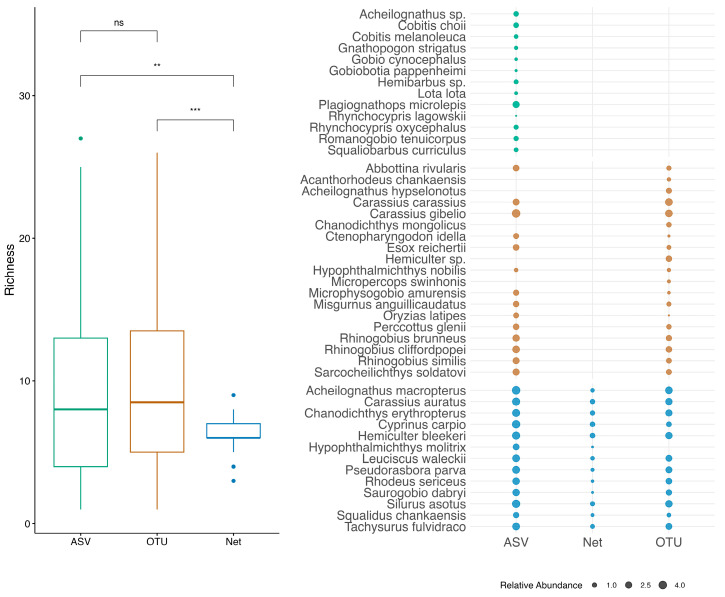
Species compositions revealed by OTU- and ASV-based eDNA survey and capture-based survey. Left plot presents alpha diversity based on species richness, asterisks indicate the statistical significance (*p* > 0.05: ns, *p* <0.05: *, *p* < 0.01: **, *p* < 0.001: ***); right plot presents log-transformed species abundance in three datasets.

**Figure 3 biology-14-01678-f003:**
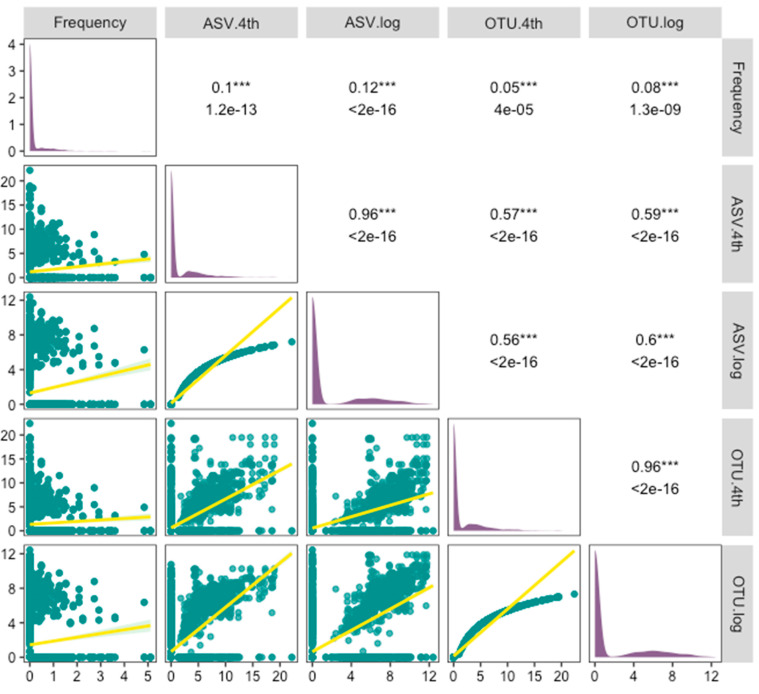
Pairwise association between the number of eDNA reads in samples (4th root- and log-transformed) and the fish frequencies in survey hauls, labelled with *r*^2^ (first line) and *p* value (second line), asterisks indicate the statistical significance (*p* < 0.001: ***).

**Figure 4 biology-14-01678-f004:**
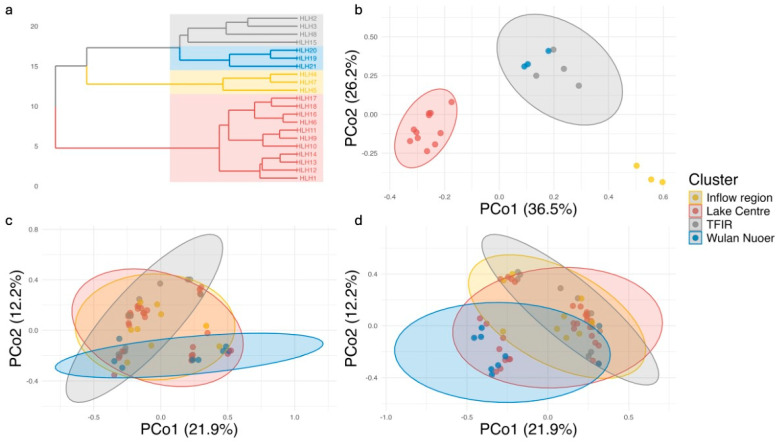
Community structure and clustering of fish assemblages in Hulun Lake based on capture-based and eDNA datasets. (**a**) Hierarchical clustering dendrogram of data in net using Ward’s method on Jaccard distances. (**b**–**d**) Principal coordinates analysis (PCoA) of capture-based (**b**), ASV-based eDNA (**c**), and OTU-based eDNA (**d**) datasets using Jaccard dissimilarity. Ellipses represent 95% confidence intervals around clusters defined from the capture-based dataset.

**Figure 5 biology-14-01678-f005:**
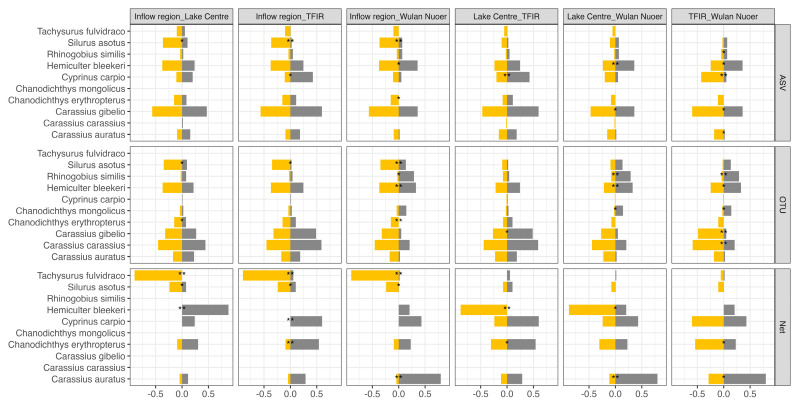
Species contributions to dissimilarity between fish community clusters using SIMPER analysis across three data types: traditional netting (Net), eDNA ASV-based, and eDNA OTU-based datasets. For each cluster pairwise comparison (columns), the five most influential taxa are shown (top three contributors in each dataset are highlighted), with positive and negative bars representing average abundance in each group. Asterisks indicate the statistical significance of contributions (*p* < 0.05: *, *p* < 0.01: **, *p* < 0.001: ***).

**Table 1 biology-14-01678-t001:** Statistical significance of differences in fish community structure among ecological groups, from netting data, eDNA ASV-based, and eDNA OTU-based datasets, as resolved using PERMANOVA.

	*Df*	*SS*	*r* ^2^	*F*	*p*
Net					
Model	3	3.5841	0.70293	13.409	**0.001**
Residual	17	1.5147	0.29707		
Total	20	5.0988	1		
OTU					
Model	3	2.3209	0.12514	2.6224	**0.001**
Residual	55	16.2255	0.87486		
Total	58	18.5464	1		
ASV					
Model	3	1.9136	0.10572	2.0885	**0.005**
Residual	53	16.1878	0.89428		
Total	56	18.1014	1		

Significant differences (*p* < 0.05) have been bolded.

**Table 2 biology-14-01678-t002:** Statistical significance of differences in fish community structure among biological groups, from netting data and OTU- and ASV-based eDNA data, as resolved using pairwise PERMANOVA.

Pairs	*SS*	*F*	R^2^	*p*
Net				
Lake Centre vs. Inflow region	1.7931209	28.711902	0.7052459	0.004
Lake Centre vs. TFIR	1.2167088	13.669433	0.5125506	0.002
Lake Centre vs. Wulan Nuoer	1.0026934	16.700379	0.5818871	0.004
Inflow region vs. TFIR	1.1072019	6.970397	0.5823029	0.032
Inflow region vs. Wulan Nuoer	1.1911925	13.325187	0.7691223	0.1
TFIR vs. Wulan Nuoer	0.4251528	2.777786	0.3571435	0.034
OTU				
Lake Centre vs. Inflow region	0.5677432	1.899191	0.0488234	0.025
Lake Centre vs. TFIR	0.5837242	2.030721	0.0494927	0.036
Lake Centre vs. Wulan Nuoer	1.0187869	2.99325	0.07484388	0.003
Inflow region vs. TFIR	0.4398811	2.179936	0.10802492	0.055
Inflow region vs. Wulan Nuoer	0.9516775	3.036197	0.15949598	0.004
TFIR vs. Wulan Nuoer	1.2599592	4.391153	0.19611108	0.001
ASV				
Lake Centre vs. Inflow region	0.5756267	1.834298	0.04848242	0.036
Lake Centre vs. TFIR	0.6427387	2.133024	0.05314884	0.02
Lake Centre vs. Wulan Nuoer	0.6415883	1.82848	0.04964853	0.045
Inflow region vs. TFIR	0.5698036	2.625317	0.12728614	0.019
Inflow region vs. Wulan Nuoer	0.5852228	1.853017	0.10995163	0.077
TFIR vs. Wulan Nuoer	0.8637256	3.00242	0.15010283	0.003

## Data Availability

All code to reproduce the analyses in this study can be obtained from 10.5281/zenodo.15831787. The raw sequencing reads generated from this study has been submitted to the Sequence Read Archive (SUB15453882).

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
