# Peer review of "Spatial Heterogeneity and Methodological Insights in Fish Community Assessment: A Case Study in Hulun Lake"

_biology, 2025, doi:10.3390/biology14121678_

Round 1

Reviewer 1 Report

Comments and Suggestions for Authors

This work investigates fish community composition in Hulun Lake using both environmental DNA (eDNA) metabarcoding and capture-based surveys. The study compares two bioinformatic approaches (ASV and OTU pipelines) to evaluate differences in species detection and community structure, and it identifies four ecological zones with distinct assemblages. Importantly, the study highlights the advantages of eDNA in detecting rare or small-bodied species, while demonstrating that capture-based surveys still provide valuable abundance data. The authors conclude that an integrated monitoring framework is optimal for conservation of Hulun Lake’s fish biodiversity.

The research is timely and relevant, offering both methodological and ecological insights. The dataset is well structured, and the analyses are generally sound. The manuscript is clear and informative, but there are some points that need clarification and revision.

  1. The study addresses ASV vs. OTU pipeline performance, but the novelty could be better emphasized. Similar comparisons have been published; the authors should clarify what is new in the context of Hulun Lake (e.g., first application in this ecosystem, or specific conservation relevance).
  2. It is recommended to clarify the specific key manifestations of the claim that "traditional net capture detects more heterogeneous patterns"; the authors' description of this point is somewhat vague.
  3. There is an obvious grammatical error in line 85: "Dispute detailed sampling and experiment protocols and careful work can constrain false positives". It is suggested to revise "Dispute" to "Despite" and "experiment" to "experimental", and further clarify the semantic expression of the sentence.
  4. Regarding the differences between ASV and OTU, the authors only cite literature in a general manner to illustrate their impacts, but fail to combine the water characteristics of Hulun Lake to explain the potential differential risks of the two bioinformatic pipelines in "accurate species identification" in this region. It is recommended to add 1-2 sentences of correlative analysis on "methodological bottlenecks - regional characteristics", thereby strengthening the necessity and innovation of this study.
  5. For Lines 203-206, does the transformed data meet the normality requirement for linear regression analysis? It is recommended to clearly supplement the verification result of this aspect.
  6. It is recommended to supplement the specific R2 values of the regression model, and briefly mention that "the residual normality test shows the transformed data meet the model assumptions (Shapiro-Wilk test, p > 0.05)" to enhance the credibility of the conclusions.
  7. It is recommended to supplement the objective limitations of this study in the Discussion section.
  8. It is recommended to unify the terminology referring to traditional net capture methods, such as "capture-based survey" and "trawl surveys".
  9. The regression results are promising (correlations between eDNA and nets), but more discussion is needed on the ecological meaning of the slope (semi-quantitative vs. fully quantitative). Suggest authors to summarize other works around this topic.
  10. Ensure all references are formatted consistently (some lack italics or full details).

Author Response

Comments 1: The study addresses ASV vs. OTU pipeline performance, but the novelty could be better emphasized. Similar comparisons have been published; the authors should clarify what is new in the context of Hulun Lake (e.g., first application in this ecosystem, or specific conservation relevance).

Response 1: We thank the reviewer for this important point. We have addressed this point by adding: While comparisons between ASV and OTU pipelines have been conducted in various aquatic ecosystems, their performance and implications for conservation have not been evaluated in the unique context of Hulun Lake.

Comments 2: It is recommended to clarify the specific key manifestations of the claim that "traditional net capture detects more heterogeneous patterns"; the authors' description of this point is somewhat vague.

Response 2: Agree, revision for clarifying this point. From both PCoA analysis and PERMANOVA model, the strongest dissimilarities (clear separation between clusters) were observed in the capture-based data, while the eDNA datasets revealed weaker but still significant contrasts (Table 2, Figure 4). Added ‘From both PCoA analysis and PERMANOVA model’ and ‘(clear separation between clusters)’.

Comments 3: There is an obvious grammatical error in line 85: "Dispute detailed sampling and experiment protocols and careful work can constrain false positives". It is suggested to revise "Dispute" to "Despite" and "experiment" to "experimental", and further clarify the semantic expression of the sentence.

Response 3: Sorry for typos and grammar errors, we have went through the whole manuscript and make corrections.

Comments 4: Regarding the differences between ASV and OTU, the authors only cite literature in a general manner to illustrate their impacts, but fail to combine the water characteristics of Hulun Lake to explain the potential differential risks of the two bioinformatic pipelines in "accurate species identification" in this region. It is recommended to add 1-2 sentences of correlative analysis on "methodological bottlenecks - regional characteristics", thereby strengthening the necessity and innovation of this study.

Response 4: Agreed, we have put a few more sentences to discuss methodological bottlenecks in Hulun Lake, focusing on hybrids and close species. (L333) At the same time, the OTU pipeline's tendency to cluster sequences with high genetic similarity likely led to the misidentification or merging of closely related species or hybrids, such as the absence of silver carp (Hypophthalmichthys molitrix) which was likely clustered with bighead carp (H. nobilis). Conversely, the ASV pipeline's single-nucleotide resolution, while powerful, may be prone to over-splitting in this high-biomass environment, potentially detecting rare variants that could be false positives. Thus, a du-al-pipeline approach, as adopted here, provides complementary perspectives.

Comments 5: For Lines 203-206, does the transformed data meet the normality requirement for linear regression analysis? It is recommended to clearly supplement the verification result of this aspect.

Response 5: Thank you, we examined residual distributions using Q–Q plots (added a plot in supporting information Figure S4) and Shapiro–Wilk tests. Some models showed deviations from strict normality. However, with the large sample size (n = 5800), linear regression is robust to such deviations according to the central limit theorem. Additionally, visual inspection confirmed that residuals were symmetrically distributed without strong skewness or heteroscedasticity, supporting the appropriateness of linear regression for our analysis.

Comments 6: It is recommended to supplement the specific R2 values of the regression model, and briefly mention that "the residual normality test shows the transformed data meet the model assumptions (Shapiro-Wilk test, p > 0.05)" to enhance the credibility of the conclusions.

Response 6: We have added Q-Q plots of each model in the supplement, hope it will enhance the credibility of our methods.

Comments 7: It is recommended to supplement the objective limitations of this study in the Discussion section.

Response 7: Agree, we have emphasized the clustering method is not the only way causing discrepancies in survey results. (L 344-346) It is worth noting that clustering methods alone may not fully explain these discrepancies, additional factors such as primer choice, PCR efficiency, and the thresholds used in bioinformatic processing also play critical roles [37,38,39,33].

Furthermore, we have added a paragraph to address methodological chanllenges. (L 401-406) When framing eDNA-based regulations, it is crucial to consider the method's lim-itations. For instance, PCR inhibition from organic matter or algal blooms, which can reduce detection sensitivity, and incomplete reference databases, which can lead to false positives/negatives or misidentification of hybrids and closely related species and so on. Future applications should focus on expanding reference databases and incorporating technical replicates to improve reproducibility[51,52,53].

Comments 8: It is recommended to unify the terminology referring to traditional net capture methods, such as "capture-based survey" and "trawl surveys".

Response 8: Thank you, we have checked the manuscript and unify the terminology of traditional capture methods.

Comments 9: The regression results are promising (correlations between eDNA and nets), but more discussion is needed on the ecological meaning of the slope (semi-quantitative vs. fully quantitative). Suggest authors to summarize other works around this topic.

Response 9: Agree, we emphasized this point by adding a sentence: (L327) The significant positive correlation also supports the consensus that eDNA metabarcoding provides semi-quantitative data[11].

Comments 10: Ensure all references are formatted consistently (some lack italics or full details).

Response 10: Thank you, we checked references all over again.

Reviewer 2 Report

Comments and Suggestions for Authors

In this study, authors compared ASV and OUT pipelines in revealing biodiversity, and also compare these methods with traditional catch survey, which is meaningful for further application of eDNA biomonitoring of local fish species. Insightful ecological outcome will be helpful on guiding conservational plans especially on spatial priorities. Overall, the manuscript is strong and suitable for publication. I recommend minor revisions to improve clarity, flow, and some technical details. 

  1. Clarify briefly why 21 sampling sites were chosen. Was this based on logistical constraints, or considered sufficient for spatial coverage? Why include Wuerxun River?
  2. The discussion of ASV vs. OTU differences is superficial. The manuscript acknowledges discrepancies but does not explore their implications in depth.
  3. This is an application of eDNA, the outcome should emphasize on potential implication for focal ecosystems and eDNA methodological improvement.

Author Response

Comments 1: Clarify briefly why 21 sampling sites were chosen. Was this based on logistical constraints, or considered sufficient for spatial coverage? Why include Wuerxun River?

Response 1: Thank you, we have added “These sampling sites allow assessment of hydrological connectivity and human impact on fish assemblages by including high human impact areas – fish farms area, low human impact areas – lake centre and the inflow areas from connected Wuerxun River” in 2.1, to explain the sampling sites designing reason

Comments 2: The discussion of ASV vs. OTU differences is superficial. The manuscript acknowledges discrepancies but does not explore their implications in depth.

Response 2: Agree, very good suggestion. We add a paragraph in 4.1 to address implications of ASV and OTU. These discrepancies between ASV- and OTU-based pipelines are not merely technical artifacts but have practical implications. ASVs, which retain single-nucleotide resolution, may over-split true biological variants, leading to apparent higher richness but potentially inflating false positives. In contrast, OTU clustering can mask fine-scale diversity by grouping closely related species, resulting in more conservative richness estimates but potentially higher concordance with capture data. This divergence indicates that pipeline choice can influence conservation conclusions, particularly regarding detection of rare or cryptic taxa. For applied monitoring, ASVs may be preferable for maximizing sensitivity, while OTUs may better support abundance correlations with conventional surveys. A dual-pipeline approach, as adopted here, provides complementary perspectives.

Comments 3: This is an application of eDNA, the outcome should emphasize on potential implication for focal ecosystems and eDNA methodological improvement.

Response 3: Agree, we have rephrased some parts and added a few sentences. (L 394-406) Therefore, we recommend prioritizing the Lake Centre zone as a core conservation area with restricted anthropogenic disturbance. Fisheries adjacent zones require regular monitoring for invasive and aquaculture related species, particularly generalists such as Carassius gibelio. Studying the ecological effect of invasive and introduced species and closely monitoring invasive and generalist species to safeguard native assemblages[50]. To support this, eDNA monitoring could be integrated into regulatory frameworks to pro-vide early warnings of invasive species spread or biodiversity loss.

When framing eDNA-based regulations, it is crucial to consider the method's limitations. For instance, PCR inhibition from organic matter or algal blooms, which can reduce detection sensitivity, and incomplete reference databases, which can lead to false positives/negatives or misidentification of hybrids and closely related species and so on. Future applications should focus on expanding reference databases and incorporating technical replicates to improve reproducibility[51,52,53].

Reviewer 3 Report

Comments and Suggestions for Authors

The manuscript "Spatial Heterogeneity and Methodological Insights in Fish Community Assessment: An eDNA and Capture-based Survey Study of Hulun Lake" presents an integrative biodiversity monitoring study in Hulun Lake, a UNESCO Biosphere Reserve under increasing anthropogenic pressure. The authors compare eDNA metacoding (using ASV and OTU pipelines) with traditional capture-based surveys, highlighting differences in species detection and community spatial structuring. The study reveals that eDNA detects 2–3 times more species, including rare and endangered taxa, while maintaining strong correlations with traditional abundance data. The article makes a timely contribution to conservation biology and provides methodological insights relevant to eDNA applications.

Overall, the manuscript is innovative, well-written, and addresses a topic of broad interest. However, some revisions are needed to improve clarity, depth, and presentation. For example:

Language and Readability: The manuscript contains several grammatical errors and inappropriate phrasing (e.g., "OUT" instead of "OTU," "Dispute detailed sampling..."). A full review in English is strongly recommended to ensure clarity and professional readability.

Discussion of methodological limitations: While the strengths of eDNA are well emphasized, limitations (e.g., PCR inhibition, false positives/negatives, incomplete reference databases, and potential contamination by aquaculture DNA) are only briefly mentioned. A more explicit discussion of these caveats is needed to provide a balanced view and guide future work.

Conservation implications: Conservation recommendations remain somewhat generic. Authors should connect their findings more directly to actionable management strategies (e.g., prioritizing core lake zones as refuges, monitoring invasive species in fisheries, or integrating eDNA into regulatory frameworks). This would increase the applied impact of the study.

Figures and tables: Several figures (e.g., Figures 2–5) are visually dense and may be difficult for readers to interpret. Consider improving legends, simplifying color schemes, and splitting panels where necessary to improve accessibility.

Summary: Correct minor inconsistencies (e.g., “ASV vs OUT” → “ASV vs OTU”).

Methods: Clarify details about negative controls (were positives detected and how were they treated?).

Results: Some results are repeated between the text and figure legends; simplify to avoid redundancy.

References: Ensure consistent formatting (some entries lack proper italics or DOI formatting).

Figures: Add clearer legends to indicate sampling locations and zones on the map (Figure 1).

Author Response

Reviewer 3:

Comments 1: Language and Readability: The manuscript contains several grammatical errors and inappropriate phrasing (e.g., "OUT" instead of "OTU," "Dispute detailed sampling..."). A full review in English is strongly recommended to ensure clarity and professional readability.

Response 1: Thank you, and we are sorry for typos and grammar mistakes. These mistakes have been corrected.

Comments 2: Discussion of methodological limitations: While the strengths of eDNA are well emphasized, limitations (e.g., PCR inhibition, false positives/negatives, incomplete reference databases, and potential contamination by aquaculture DNA) are only briefly mentioned. A more explicit discussion of these caveats is needed to provide a balanced view and guide future work.

Response 2: Agree, we add a paragraph to address methodological limitations after L400.  When framing eDNA-based regulations, it is crucial to consider the method's limitations. For instance, PCR inhibition from organic matter or algal blooms, which can reduce detection sensitivity, and incomplete reference databases, which can lead to false positives/negatives or misidentification of hybrids and closely related species and so on. Future applications should focus on expanding reference databases and incorporating technical replicates to improve reproducibility[51,52,53].

Comments 3: Conservation implications: Conservation recommendations remain somewhat generic. Authors should connect their findings more directly to actionable management strategies (e.g., prioritizing core lake zones as refuges, monitoring invasive species in fisheries, or integrating eDNA into regulatory frameworks). This would increase the applied impact of the study.

Response 3: Thank you, we agree with your opinion and add more discussion in conservation implications by the end of first paragraph in 4.3. Therefore, we recommend prioritizing the Lake Centre zone as a core conservation area with restricted anthropogenic disturbance. Fisheries adjacent zones require regular monitoring for invasive and aquaculture related species, particularly generalists such as Carassius gibelio. Studying the ecological effect of invasive and introduced species and closely monitoring invasive and generalist species to safeguard native assemblages[50]. To support this, eDNA monitoring could be integrated into regulatory frameworks to pro-vide early warnings of invasive species spread or biodiversity loss..

Comments 4: Figures and tables: Several figures (e.g., Figures 2–5) are visually dense and may be difficult for readers to interpret. Consider improving legends, simplifying color schemes, and splitting panels where necessary to improve accessibility.

Response 4: Agree, we have made a few changes. First, we enlarge and font labels in figures. Secondly, we clarify captions of figures. Hope they will make the figures easier for interpretation, if you have more specific suggestions, we will work on it accordingly. Thirdly, we also changed color panel in figure 2 and figure 4 for better visualization.

Comments 5: Summary: Correct minor inconsistencies (e.g., “ASV vs OUT” → “ASV vs OTU”).

Response 5: Thank you, they have been revised.

Comments 6: Methods: Clarify details about negative controls (were positives detected and how were they treated?).

Response 6: Thank you, we add a paragraph to describe how we set thresholds based on negative controls. Reads present in negative controls were used as species-specific cutoffs, and any taxon with fewer reads than observed in negative controls was excluded from downstream analyses. This procedure ensured that potential contamination did not affect community structure results.

Comments 7: Results: Some results are repeated between the text and figure legends; simplify to avoid redundancy.

Response 7: thank you. We checked figures and texts, there are some redundancy. P values were removed from text in following sentences: Linear regression analysis revealed highly significant associations  between species abundance metrics across capture-based and eDNA approaches (Figure 3). A significant positive correlation was observed between frequencies from nets and both the ASVs and OTUs reads under both log and 4th-root transformations. The OTU results demonstrated a slightly better fit than the ASV results in these models, likely due to a less proportion of absent species. Strong correlations were also observed between the two eDNA pipelines under both transformation methods, indicating similar performance and supporting the consistency of the alpha diversity comes.

Figure 4 caption: revealing four distinct spatial clusters: Lake Centre, Inflow region, Fisheries, and Wulan Nuoer. -> deleted

Comments 8: References: Ensure consistent formatting (some entries lack proper italics or DOI formatting).

Response 8: Thank you, we make a few changes accordingly. Since DOI is not mandatory and we didn’t include DOI for majority references, we decide to go without them.

Comments 9: Figures: Add clearer legends to indicate sampling locations and zones on the map (Figure 1).

Response 9: Agree. We labelled fishery zone and inflow region, but Wulunnuoer lake is already separated from other so we leave it being there.  And lake center contains the rest sampling sites, which is difficult to label, we prefer to keep it unlabeled from aesthetic concerns.

Reviewer 4 Report

Comments and Suggestions for Authors

Letter to Authors
biology-3885644-v1
Spatial Heterogeneity and Methodological Insights in Fish Community Assessment: An eDNA and Capture-based Survey Study of Hulun Lake
Zifang Liu, Yuetong Zhang, Yanan Pan, Zhousunxi Ma, Xin Han, Ziqi Zhou, Shuang Tian, Bingjiao Sun

250920

Dear authors,
Your potentially interesting MS needs at least a round of substantial revision. Several inconsistencies make it difficult to see the content. See below for detail. 

L14 simple summary
As the author guide says, it is not a shorter version of the abstract. It works for a news feed for media. When writing it, imagine if you were a media journalist who do not like to read long stories in limited working times. Journalists like to read short stories in a veni-vidi-vici style. When they are interested in your paper seeing your flash presentation, they will read further or call you to gather stories to write.
See examples at:
http://doi.org/10.3390/biology14091278
http://ddoi.org/10.3390/biology14091266
Consult with a public relation officer of your institution.

L16
we need accurate approach to monitor focal species. In this study -> delete (-11 words)

L19
We also tested two different bioinformatic processes to analyze the DNA data. -> delete (-12 words)

L20
We found that -> delete (-3 words)

L22
such as near fisheries, river inflows, and the open lake -> delete (-10 words)

L23
it is less disturbed by human activities -> of reduced human disturbance (-3 words)

L24-26
Our results show that using eDNA together with traditional nets provides the best picture of fish diversity. This combined approach can help guide better conservation efforts to protect the lake's ecosystem. -> delete either of these two sentences (-17 or -14 words)

L34
Hulun Lake -> the Hulun Lake
What Hulun Lake has been specified in L28. Check thoroughly. 

L40
the Fisheries area (in-house terminology) -> areas near fish farms
Use of this term in the main text is fine after specifying it explicitly in L190. 

L43
too, notably, -> , whereas
nets -> net

L58
and provider of -> providing
services, acting -> services and acting

L62
Hulun Lake .. -> new paragaraph

L68
water levels have declined -> water levels have declined and quality has decayed
Overfishing and aquaculture seem not affect water level of the lake, but water quality. 

L84
ASV vs. OTU -> Amplicon Sequence Variant (ASV) vs. Operational Taxonomic Unit (OTU)
Spell-out at the first place independent from abstract section. 

L104
three biological replicates were taken and for each biological replicate, 1 L of surface water was filtered concentrate eDNA ??
Scheme of the replication is unclear. Did you sampled 3L of surface water into a bottle at each site and aliquoted into three? Did you sampled 1L into a bottle and repeated three times at the site?

L107
was taken into the field ??
Did you bring a bottle containing DW to the lake and do something (what?) on it before bringing back to the laboratory? 

L112
Manual for the Survey of Inland Water Fishery Natural Resources
Reference needed. 

L116
12 types of mesh sizes
Give at least minimum and maximum mesh sizes, or all. This may be relevant for method comparison. Why small fish was absent from net samples? See sub-section 4.1. 

L146
prepared and -> delete

L150
pipelines -> pipelines: 
based on two different ideologies, (does not make sense) -> delete

L151
amplicon sequence variants (ASVs) -> ASVs

L154
QIIME2 ASV-Based Workflow: (not a complete sentence) -> QIIME2 ASV-Based Workflow consisted [the following] two main steps:

L160
is following -> followed

L161
and the main steps are briefly present here: -> with [the following] four main steps:

L162
'obigrep'. -> 'obigrep';

L164
'obimultiplex'. -> 'obimultiplex';

L165
'obiuniq'. -> 'obiuniq';

L171
OTUs and ASVs ?? -> ASVs and OTUs ?
Make the order of description consistent throughout the text. Do not disturb readers' short term memory for reading. I propose to follow the order in L92,150-170, but when you like to inverse, so you should mention in your preferred order from the beginning to the end. 

L174
Last, (verbose) -> delete
smaller -> fewer

L175
lower the chances of -> eliminate

L176
OBITools and QIIME2 -> QIIME2 and OBITools

L188-194
The optimal number of clusters .. of Hulun Lake. -> revise
Option-1
-> move to the result section
Option-2
Give location IDs for four clusters (Fisheries, Inflow, Wulan Nuoer, Lake center). 

L195
biological (does not make sense) -> the four location

L198
PERMANOVA -> permutational multivariate ANOVA (PERMANOVA)
"ANOVA" is marginally OK, but PERMANOVA is not as familiar as ANOVA. 

L201
SIMPER -> similarity percentile (SIMPER)

L202
four ecological groups ?? -> the four location clusters ?
What do "ecological groups" mean? Jumping argument?

L210
2,190,909 reads (OBITools/OTU) and 2,892,001 reads (QIIME2/ASV
See my comment on L171. 

L219
Cite Figure 2 over here. Put essential information (what figure) forward. 
Top-heavy documents are preferable, because readers can stop reading anywhere getting the best information at that point. Check thoroughly for other similar cases citing figures and tables. 

L241,296,303 figure pictures
Fonts in the pictures are too small to see. They should at least as large as those in the main text. To do it, enlarge fonts only. Do not make fool enlargement of the entire pictures. 

L234
Cite Figure 3 over here. 

L242
OUT and ASV -> ASV and OTU

L250
ecological groups which are described in method section -> groups of locations which seem to have ecological bases
This revision corresponds to the option-2 of revising L188-194. A lower part (which ..) fills a gap in a jumping argument, and hereafter you may use "ecological groups". 
L251
4 -> four
Spell-out numbers equal to or below ten. 

L252
Figure 3 ?? -> Figure 4 ?

L273
goldfish ??
See L223. 

L276,290,291,368
fisheries -> Fisheries (in Italics)
See L190. 

L277,368
inflow -> Inflow (in Italics)

L278,365
lake centre -> Lake Centre (in Italics)

L318
validate -> validated

L374
conclusions of Dos Santos & Blabolil[47] -> previous conclusion [47]
A merit of numbered citation is to save readers' short term memory spaces when reading. Readers can go straightforward through the story-flow without outflow of author names [and published years]. 

L385,393
Fisheries -> in Italics

L386
Lake Centre -> in Italics

L394-397
Studying the ecological effect .. across studies[53,54,55]. (she2 zu2) -> delete

L398
nuanced (weak) -> effective
Hulun Lake's -> freshwater lake (general conclusion)

L400 conclusions
Delete. You have well presented a conclusion in the previous paragraph. A conclusion section is not mandatory for this journal. 

L415 supplements
File "supporting_information.docx" and "supporting_information.xlsx" give different figures and tables. 

L445
Key -> in lower or title case

L446
The -> the

L448
"Not applicable". -> Not applicable.
Do not copy-and-paste a template. 

L455
Declare .. the results". -> None declared. 
Do not copy-and-paste template examples. 

L463 abbreviations
Delete. Do not copy-and-paste examples. 

L465 references
Check the reference list carefully again from the beginning. Reference lists are frequently hotbeds of errors. You might add, omit or swap citation in the main text on the way internal revision. Numbering of the references might then shift. If so, readers think you are making irrelevant citation. It is the authors' responsibility that all references are properly cited.

Check thoroughly to make sure:
if separators of authors are a colon (L507,etc),
if surnames come first with a comma followed by initials of middle and given names with dots for all authors (L507,etc),
if all authors are listed except for >10 in "et al." (L595),
if paper titles are in lower case (L470,etc),
  The journal does not rule case usage, but follow your majority.  
if scientific names are in Italics (L535,etc),
if journal titles are abbreviated when possible (L468,etc many),
if spelled-out journal titles do not accompany a dot nor a comma (L468,etc all),
if book/journal titles are in Italic title case (L467,515,etc all),
if published year comes next to journal/book title in bold Italics (467,etc all),
etc.
See the citation guide at:
https://www.mdpi.com/authors/references/

L540
?. -> ?

L544
See L547. 

L578
?

Supporting Information Table S2
Make 2D matrix with species in a line and locations in a row. Line and row can be inverted when you like. 

Supporting Information Table S3
Swap OBITools and QIIME2 columns. See my comment on L171. 

Figure S2
Swap OTU and ASV pictures. See my comment on L171. 

Comments on the Quality of English Language

Several incomplete sentences exist. 

Author Response

Comment 1: simple summary

As the author guide says, it is not a shorter version of the abstract. It works for a news feed for media. When writing it, imagine if you were a media journalist who do not like to read long stories in limited working times. Journalists like to read short stories in a veni-vidi-vici style. When they are interested in your paper seeing your flash presentation, they will read further or call you to gather stories to write.

See examples at:

http://doi.org/10.3390/biology14091278

http://ddoi.org/10.3390/biology14091266

Consult with a public relation officer of your institution.

L16

we need accurate approach to monitor focal species. In this study -> delete (-11 words)

L19

We also tested two different bioinformatic processes to analyze the DNA data. -> delete (-12 words)

L20

We found that -> delete (-3 words)

L22

such as near fisheries, river inflows, and the open lake -> delete (-10 words)

L23

it is less disturbed by human activities -> of reduced human disturbance (-3 words)

L24-26

Our results show that using eDNA together with traditional nets provides the best picture of fish diversity. This combined approach can help guide better conservation efforts to protect the lake's ecosystem. -> delete either of these two sentences (-17 or -14 words)

Response 1: Thank you for your suggestion. We agree with your opinion. Therefore, we rewrote the simple summary, as following: Hulun Lake, an important nature reserve in China, is facing threats from climate change and human activities. To protect its biodiversity, we compared a molecular meth-od—analyzing DNA left by organisms in the water (environmental DNA)—with the capture-based traditional survey methods. The eDNA method detected 2–3 times more fish species than by nets, including rare and endangered species. Both methods showed that fish communities differ in different parts of the lake. The central lake area had a unique assemblage of species, likely because of reduced human disturbance. Our results show that using eDNA together with traditional nets can help guide more effective conservation efforts to protect the lake’s ecosystem.

Comment 2: L34

Hulun Lake -> the Hulun Lake

What Hulun Lake has been specified in L28. Check thoroughly.

Response 8: Thank you, but the usage of ‘Hulun Lake’ is a proper noun like ‘Lake Victoria’ and it stands alone without ‘the’. It also consistent with other research articles, like https://doi.org/10.3389/fmicb.2025.1550555, https://doi.org/10.1016/j.proenv.2012.01.103 and etc. 

Comment 3: L40

the Fisheries area (in-house terminology) -> areas near fish farms

Use of this term in the main text is fine after specifying it explicitly in L190.

L43

too, notably, -> , whereas

nets -> net

L58

and provider of -> providing

services, acting -> services and acting

L62

Hulun Lake .. -> new paragaraph

L68

water levels have declined -> water levels have declined and quality has decayed

Overfishing and aquaculture seem not affect water level of the lake, but water quality.

L84

ASV vs. OTU -> Amplicon Sequence Variant (ASV) vs. Operational Taxonomic Unit (OTU)

Spell-out at the first place independent from abstract section.

Response 3: Agree, these places have been revised. Thank you for your suggestion.

Comment 4: L104

three biological replicates were taken and for each biological replicate, 1 L of surface water was filtered concentrate eDNA ??

Scheme of the replication is unclear. Did you sampled 3L of surface water into a bottle at each site and aliquoted into three? Did you sampled 1L into a bottle and repeated three times at the site?

Response 4: Agree, we sampled 1L surface water for 3 time, for better understanding, the revised version is: From each sampling site, three biological replicates were taken separately and for each biological replicate, 1 L of surface water was filtered to concentrate eDNA.

Comment 5: L107

was taken into the field ??

Did you bring a bottle containing DW to the lake and do something (what?) on it before bringing back to the laboratory?

Response 5: We brought a bottle of distilled water with us in the car and the boat we used in sampling, where the DW can be exposed to our sampling equipment. Then we brought it back to the lab, and filtered the water with a Sterivex in the same lab. We rephrased the sentence: 1 L of distilled water was taken into the field as a field negative control by exposing to sampling equipment. At least one negative control was prepared per sampling day and treated identically to the other samples.

Comment 6: L112

Manual for the Survey of Inland Water Fishery Natural Resources

Reference needed.

Response 6: Agree, we were using the method used in a paper which is cited now. So that this line is rephrased as: Fish samples were collected using gill netting at same sampling locations as eDNA sampling, mainly following process[16]: prior to sampling with gillnets, fishing permits were obtained from relevant authorities following local conservative measures. The fishing gear used in this survey, included the multi-mesh type gill net (2 m×30 m, composed of 12 types of mesh sizes, with mesh size ranging from 2.7 cm to 16 cm), the three-layer gill net (1 m×50 m, with mesh sizes of 3.5 cm, 5.5 cm, and 7.5 cm), and the bottom fixed trap net (0.5 m×0.5 m×15 m).

Comment 7: L116

12 types of mesh sizes

Give at least minimum and maximum mesh sizes, or all. This may be relevant for method comparison. Why small fish was absent from net samples? See sub-section 4.1.

Response 7: Agree. We added description: The fishing gear used in this survey, included the multi-mesh type gill net (2 m×30 m, composed of 12 types of mesh sizes, with mesh size ranging from 2.7 cm to 16 cm), the three-layer gill net (1 m×50 m, with mesh sizes of 3.5 cm, 5.5 cm, and 7.5 cm), and the bottom fixed trap net (0.5 m×0.5 m×15 m).

Comment 8: L146

prepared and -> delete

L150

pipelines -> pipelines:

based on two different ideologies, (does not make sense) -> delete

L151

amplicon sequence variants (ASVs) -> ASVs

L154

QIIME2 ASV-Based Workflow: (not a complete sentence) -> QIIME2 ASV-Based Workflow consisted [the following] two main steps:

L160

is following -> followed

L161

and the main steps are briefly present here: -> with [the following] four main steps:

L162

'obigrep'. -> 'obigrep';

L164

'obimultiplex'. -> 'obimultiplex';

Response 26:

L165

'obiuniq'. -> 'obiuniq';

L171

OTUs and ASVs ?? -> ASVs and OTUs ?

Make the order of description consistent throughout the text. Do not disturb readers' short term memory for reading. I propose to follow the order in L92,150-170, but when you like to inverse, so you should mention in your preferred order from the beginning to the end.

L174

Last, (verbose) -> delete

smaller -> fewer

L175

lower the chances of -> eliminate

L176

OBITools and QIIME2 -> QIIME2 and OBITools

Response 8: Agree, sentences were revised accordingly.

Comments 9:

L188-194

The optimal number of clusters .. of Hulun Lake. -> revise

Option-1

-> move to the result section

Option-2

Give location IDs for four clusters (Fisheries, Inflow, Wulan Nuoer, Lake center).

Response 9: Agree, this part has been moved to result section.

Comment 10: L195

biological (does not make sense) -> the four location

L198

PERMANOVA -> permutational multivariate ANOVA (PERMANOVA)

"ANOVA" is marginally OK, but PERMANOVA is not as familiar as ANOVA.

L201

SIMPER -> similarity percentile (SIMPER)

L202

four ecological groups ?? -> the four location clusters ?

What do "ecological groups" mean? Jumping argument?

L210

2,190,909 reads (OBITools/OTU) and 2,892,001 reads (QIIME2/ASV

See my comment on L171.

L219

Cite Figure 2 over here. Put essential information (what figure) forward.

Top-heavy documents are preferable, because readers can stop reading anywhere getting the best information at that point. Check thoroughly for other similar cases citing figures and tables.

Response 10: Agree, sentences has been changed accordingly. Figure citation also revised thoroughly.

Comment 11: L241,296,303 figure pictures

Fonts in the pictures are too small to see. They should at least as large as those in the main text. To do it, enlarge fonts only. Do not make fool enlargement of the entire pictures.

Response 11: Thank you, we have enlarged texts in these figures.

Comment 12: L234

Cite Figure 3 over here.

L242

OUT and ASV -> ASV and OTU

Responses 12: Agree, revised accordingly.

Comments 13: L250

ecological groups which are described in method section -> groups of locations which seem to have ecological bases

This revision corresponds to the option-2 of revising L188-194. A lower part (which ..) fills a gap in a jumping argument, and hereafter you may use "ecological groups".

Response 13: Thank you, we agree with your suggestion, we move the L188-194 part to result section and changed this line accordingly.

Comment 14: L251

4 -> four

Spell-out numbers equal to or below ten.

L252

Figure 3 ?? -> Figure 4 ?

L273

goldfish ??

See L223.

L276,290,291,368

fisheries -> Fisheries (in Italics)

See L190.

L277,368

inflow -> Inflow (in Italics)

L278,365

lake centre -> Lake Centre (in Italics)

L318

validate -> validated

Comment 50: L374

conclusions of Dos Santos & Blabolil[47] -> previous conclusion [47]

A merit of numbered citation is to save readers' short term memory spaces when reading. Readers can go straightforward through the story-flow without outflow of author names [and published years].

L385,393

Fisheries -> in Italics

L386

Lake Centre -> in Italics

Response 14: Agree, content has been revised accordingly.

Comments 15: L394-397

Studying the ecological effect .. across studies[53,54,55]. (she2 zu2) -> delete

Responses: Sorry, I think there is a typo here in your comment. What do you mean?

Comment 16: L398

nuanced (weak) -> effective

Hulun Lake's -> freshwater lake (general conclusion)

Response 16: Agree, used “effective” instead.

Comment 17: L400 conclusions

Delete. You have well presented a conclusion in the previous paragraph. A conclusion section is not mandatory for this journal.

Response 17: Ok, we have removed the Conclusion section for now.

Comment 18: L415 supplements

File "supporting_information.docx" and "supporting_information.xlsx" give different figures and tables.

Response 18: Sorry, the description is provided wrongly, the correct description is provided.

Comment 19: L445

Key -> in lower or title case

L446

The -> the

L448

"Not applicable". -> Not applicable.

Do not copy-and-paste a template.

Comment 60: L455

Declare .. the results". -> None declared.

Do not copy-and-paste template examples.

Comment 61: L463 abbreviations

Delete. Do not copy-and-paste examples.

Response 19: Thank you, we have revised these errors.

Comment 20: L465 references

Check the reference list carefully again from the beginning. Reference lists are frequently hotbeds of errors. You might add, omit or swap citation in the main text on the way internal revision. Numbering of the references might then shift. If so, readers think you are making irrelevant citation. It is the authors' responsibility that all references are properly cited.

Check thoroughly to make sure:

if separators of authors are a colon (L507,etc),

if surnames come first with a comma followed by initials of middle and given names with dots for all authors (L507,etc),

if all authors are listed except for >10 in "et al." (L595),

if paper titles are in lower case (L470,etc),

  The journal does not rule case usage, but follow your majority. 

if scientific names are in Italics (L535,etc),

if journal titles are abbreviated when possible (L468,etc many),

if spelled-out journal titles do not accompany a dot nor a comma (L468,etc all),

if book/journal titles are in Italic title case (L467,515,etc all),

if published year comes next to journal/book title in bold Italics (467,etc all),

etc.

See the citation guide at:

https://www.mdpi.com/authors/references/

L540

?. -> ?

L544

See L547.

L578

?

Response 20: Sorry for these errors, we have corrected them.

Comments 21: Supporting Information Table S2

Make 2D matrix with species in a line and locations in a row. Line and row can be inverted when you like.

Responses 21: Changed

Comments 22:

Supporting Information Table S3

Swap OBITools and QIIME2 columns. See my comment on L171.

Figure S2

Swap OTU and ASV pictures. See my comment on L171.

Responses 22: Thank you, it has been changed.

Round 2

Reviewer 4 Report

Comments and Suggestions for Authors

Letter to Authors
biology-3885644-v2
Spatial Heterogeneity and Methodological Insights in Fish Community Assessment: An eDNA and Capture-based Survey Study of Hulun Lake
Zifang Liu, Yuetong Zhang, Yanan Pan, Zhousunxi Ma, Xin Han, Ziqi Zhou, Shuang Tian, Bingjiao Sun

251011

Dear authors,
I am happy to see your revised MS yet partly to some extent. Your v2 MS needs one more round of substantial revision. See below for detail. 

L14 simple summary
Your revision is not enough, but your current delivery cleared what to be omitted. Retain only the essence of essence. Combine matters of redundancy.

L15
Hulun Lake, an important nature reserve in China, is facing threats from climate change and human activities. -> delete

L16
To protect its biodiversity, we need accurate approach to monitor focal species. In this study,-> To protect biodiversity of Hulun Lake, an important nature reserve in China, 
This compact wording presents the problem and your purpose. 

L19
We also tested two different bioinformatic processes to analyze the DNA data. -> delete

L22-24
, such as .. by human activities.-> delete
Do not give your results here. When readers like to know the outline, they may read the abstract. 

L24
Our results show that using eDNA together with traditional nets provides the best picture of fish diversity. This combined approach -> The combined approach of [both] eDNA and capture survey
This compact wording presents the implication of your study. Square bracketed word can be omitted. 

L100-103
These sampling sites allow .. areas from connected Wuerxun River. -> These sampling sites, specified as four {categories, clusters}, allow assessment of hydrological connectivity and human impact on fish assemblages by including high human impact areas - fish farms area (<i>Fisheries</i>), low human impact areas - lake centre (<i>Lake Centre</i>), the inflow (<i>Inflow</i>) areas from connected Wuerxun River, and Wulan Nuoer Lake (<i>Wulan Nuoer</i>).
Specify the four categories clearly here. Use these four in-house terms afterwards throughout. Check thoroughly. Consistent use of terms will save readers' short-term memory spaces. Words in wavy braces indicate options. 

L108
one negative control ? -> one negative control (distilled water? deionized water? nanopure water? milli-Q water?) 

L113
process[16]: -> a previous report [16]. (break sentence here)
If no break, the process has only one step (permission from an authority). You might then be tempted to tie items with semicolons, but it makes a very very long sentence exhausting readers' short term memory. 
prior to sampling with gillnets, fishing permits were obtained from relevant authorities following local conservative measures. -> move to L429
If the authority provided a signed document to you, you may present the issue ID. 

L115
multi-mesh type -> {multi-mesh, multi-meshed}
Avoid repeated use of "type". See L116. 

L116
with mesh size (verbose) -> delete

L122 figure picture
fishery -> fish farm

L245
which are named with their location characters, as following (verbose) -> delete
You have already named these clusters in M&M section. 

L246
aquaculture facilities -> fish farms
Avoid exhaustion of readers' short-term memory introducing new words one after another. 

L249
Center ? -> Centre
Either will do, but use it consistently. 

L259
Wulan Nuoer Lake -> <i>Wulan Nuoer</i>
Check afterwards thoroughly. 

L289
Fisheries -> in Italics
Check afterwards thoroughly. 

L438 references
Check the reference list carefully again from the beginning. Your reference list is still a den of errors. You added, omitted or swapped citation in the main text on the way revision. Numbering of the references might then shift. If so, readers think you are making irrelevant citation. It is the authors' responsibility that all references are properly cited. Tell the truth in our reply letter. 

Check thoroughly to make sure:
if paper titles are not sandwiched by double quotes (L478,etc),
if paper titles are in lower case (L444,etc),
(the journal does not have a rule of case usage, but follow your majority)
if scientific names are in Italics (L507,etc),
if journal titles are abbreviated when possible (L442,etc all),
(when you do not know how to do it, consult with the web-of-science journal info)
if abbreviated journal title words accompany a dot,
if spelled-out journal titles do not accompany a dot nor a comma (L461,etc all),
if book titles are in Italic title case (L480,etc),
if published year comes next to journal/book title in bold Italics after insertion of a white space (L484,etc all),
(when you do not know what are bold face and white space, consult with wikipedia)
etc.
See our citation guide at:
https://www.mdpi.com/authors/references/

L519
PLOS -> PLoS

L530,etc
What is [J]?
Did you put items given by a reference handling software?

Comments on the Quality of English Language

Omit redundancy. 

Author Response

Commet 1:

L14 simple summary

Your revision is not enough, but your current delivery cleared what to be omitted. Retain only the essence of essence. Combine matters of redundancy.

L15

Hulun Lake, an important nature reserve in China, is facing threats from climate change and human activities. -> delete

L16

To protect its biodiversity, we need accurate approach to monitor focal species. In this study,-> To protect biodiversity of Hulun Lake, an important nature reserve in China,

This compact wording presents the problem and your purpose.

L19

We also tested two different bioinformatic processes to analyze the DNA data. -> delete

L22-24

, such as .. by human activities.-> delete

Do not give your results here. When readers like to know the outline, they may read the abstract.

L24

Our results show that using eDNA together with traditional nets provides the best picture of fish diversity. This combined approach -> The combined approach of [both] eDNA and capture survey

This compact wording presents the implication of your study. Square bracketed word can be omitted.

Response 1: We sincerely appreciate your comments. We made changed based on your suggestions. The new simple summary is: To protect the biodiversity of Hunlun Lake, an important nature reserve in China, we compared a molecular method—analyzing DNA left by organisms in the water (envi-ronmental DNA)—with the capture-based traditional survey methods. The eDNA method detected 2–3 times more fish species than by nets, including rare and endangered species. Both methods showed that fish communities differ in different parts of the lake. The central lake area had a unique assemblage of species. The combined approach of eDNA and capture survey can help guide more effective conservation efforts to protect the lake’s ecosystem.

Comments 2:

L100-103

These sampling sites allow .. areas from connected Wuerxun River. -> These sampling sites, specified as four {categories, clusters}, allow assessment of hydrological connectivity and human impact on fish assemblages by including high human impact areas - fish farms area (<i>Fisheries</i>), low human impact areas - lake centre (<i>Lake Centre</i>), the inflow (<i>Inflow</i>) areas from connected Wuerxun River, and Wulan Nuoer Lake (<i>Wulan Nuoer</i>).

Specify the four categories clearly here. Use these four in-house terms afterwards throughout. Check thoroughly. Consistent use of terms will save readers' short-term memory spaces. Words in wavy braces indicate options.

Response 2: Thank you, very nice solution. The revised paragraph is: These sampling sites, specified as four clusters, allow assessment of hydrological con-nectivity and human impact on fish assemblages by including high human impact areas – fish farms area (Fisheries), low human impact areas – lake centre (Lake Centre) and the inflow (Inflow) areas from connected Wuerxun River, and Wulan Nuoer Lake (Wulan Nuoer).

Comments 3:

L108

one negative control ? -> one negative control (distilled water? deionized water? nanopure water? milli-Q water?)

Response 3: Thank you for pointing it out. For clarify it was distilled water, we revised as: 1 L of distilled water was taken into the field as a field negative control by exposing to sampling equipment. At least one field negative control was prepared per sampling day and treated identically to the other samples.

Comments 4:

L113

process[16]: -> a previous report [16]. (break sentence here)

If no break, the process has only one step (permission from an authority). You might then be tempted to tie items with semicolons, but it makes a very very long sentence exhausting readers' short term memory.

prior to sampling with gillnets, fishing permits were obtained from relevant authorities following local conservative measures. -> move to L429

If the authority provided a signed document to you, you may present the issue ID.

Response 4: Agree. We shortened the sentence as suggested but believe it is appropriate to clarify technical approval in the Methods section. Therefore, we retained the sentence “Prior to sampling with gillnets, fishing permits were obtained from relevant authorities following local conservation measures” at its original location (L429). The sampling sites were located in public, non-protected waters. For non-commercial, non-destructive scientific research, only the sampling range and time are required to be reported to the relevant authorities in advance, and no specific license is issued. To make it slightly clearer, we revised the sentence: Prior to sampling with gillnets, the sampling range and time were reported to the relevant authorities in accordance with local conservation measures.

Comments 5:

L115

multi-mesh type -> {multi-mesh, multi-meshed}

Avoid repeated use of "type". See L116.

L116

with mesh size (verbose) -> delete

Response 5: Thank you, we have revised them.

Comments 6:

L122 figure picture

fishery -> fish farm

Response 6: Agree, very well spotted, we changed the label.

Comments 7:

L245

which are named with their location characters, as following (verbose) -> delete

You have already named these clusters in M&M section.

Response 7: Agree, this sentence is delected. This analysis delineated four distinct ecological groups: Fisheries – four sampling sites located adjacent to fish farms; Inflow region – 3 sites situated next to Wuerxun River inflow; Wulan Nuoer – 3 sites within Wulan Nuoer Lake which is hydrologically connected to Hulun Lake via the Wuerxun River; and Lake Center – 11 sites situated in the central basin of Hulun Lake.

Comments 8:

L246

aquaculture facilities -> fish farms

Avoid exhaustion of readers' short-term memory introducing new words one after another.

Response 8: Agree.

Comments 9:

L249

Center ? -> Centre

Either will do, but use it consistently.

Response 9: Sorry, error fixed.

Comments 10:

L259

Wulan Nuoer Lake -> <i>Wulan Nuoer</i>

Check afterwards thoroughly.

Response 10: Agree, made a few changed when mention the lake as the cluster.

Comments 11:

L289

Fisheries -> in Italics

Check afterwards thoroughly.

Response 11: Thank you, changes made.

Comments 12:

L438 references

Check the reference list carefully again from the beginning. Your reference list is still a den of errors. You added, omitted or swapped citation in the main text on the way revision. Numbering of the references might then shift. If so, readers think you are making irrelevant citation. It is the authors' responsibility that all references are properly cited. Tell the truth in our reply letter.

Check thoroughly to make sure:

if paper titles are not sandwiched by double quotes (L478,etc),

if paper titles are in lower case (L444,etc),

(the journal does not have a rule of case usage, but follow your majority)

if scientific names are in Italics (L507,etc),

if journal titles are abbreviated when possible (L442,etc all),

(when you do not know how to do it, consult with the web-of-science journal info)

if abbreviated journal title words accompany a dot,

if spelled-out journal titles do not accompany a dot nor a comma (L461,etc all),

if book titles are in Italic title case (L480,etc),

if published year comes next to journal/book title in bold Italics after insertion of a white space (L484,etc all),

(when you do not know what are bold face and white space, consult with wikipedia)

etc.

See our citation guide at:

https://www.mdpi.com/authors/references/

L519

PLOS -> PLoS

L530,etc

What is [J]?

Did you put items given by a reference handling software?

Response 12:

We sincerely appreciate your comments and apologize for any difficulties the references may have caused. All cited sources have been systematically revised according to your suggestions, and we are truly grateful for your guidance.